# Hydrophobic Epoxy Caged Silsesquioxane Film (EP-POSS): Synthesis and Performance Characterization

**DOI:** 10.3390/nano11020472

**Published:** 2021-02-12

**Authors:** Yanhong Fang, Ping Wang, Lifang Sun, Linhong Wang

**Affiliations:** 1State Key Laboratory of Power Transmission Equipment & System Security and New Technology, Chongqing University, Chongqing 400044, China; 20181101005@cqu.edu.cn; 2School of Electrical Engineering, Chongqing University, Chongqing 400044, China; 3Department of Pharmacy, Sichuan Health and Rehabilitation Vocational College, Zigong 643000, China; sunlifang0524@126.com; 4School of Smart Health, Chongqing College of Electronic Engineering, Chongqing 400013, China; 201018044@cqcet.edu.cn

**Keywords:** POSS, epoxy resin, nanocomposite, hydrophobicity, solar panel

## Abstract

Hydrophobic films are widely used in aerospace, military weapons, high-rise building exterior glass, and non-destructive pipeline transportation due to their antifouling and self-cleaning properties. This paper details the successful preparation of hydrophobic epoxy caged sesquioxane (EP-POSS) via two steps of simple organic synthesis, along with studies on the effects of viscosity and reaction time on the reaction. Interestingly, the EP-POSS presented a large contact angle of 125°, indicating its excellent hydrophobicity. The surface micromorphology was observed via FE-SEM (field emission scanning electron microscopy), transmission electron microscopy (TEM), and atomic force microscopy (AFM), and the structural composition and elemental contents were analyzed via X-ray photoelectron spectroscopy (XPS) and energy-dispersive spectrometry (EDS). Thermogravimetric analysis (TGA) and differential scanning calorimetry (DSC) tests showed that EP-POSS had excellent thermal properties, and the first degradation reaction occurred at 354 °C. The mechanical performance and abrasion resistance results demonstrated that EP-POSS could be used in solar panels.

## 1. Introduction

Polyhedral oligomeric silsesquioxane (POSS), suitable for preparing nano-level hybrid materials [1,2,3,4], is an organic–inorganic hybrid material with small size effect, excellent mechanical strength, low dielectric constant, and good compatibility [5,6,7]. POSS has been widely studied by scholars in the field of nanocomposite materials.

Recently, application prospects for POSS have been found in liquid crystal materials, ionic liquids, electronic materials, and other fields [8,9,10]. EP-POSS is an epoxy resin modified with POSS; its preparation methods include physical blending and chemical copolymerization [11,12,13]. The physical blending method refers to mixing directly and adding different proportions of POSS into epoxy resin. The incorporation of POSS into epoxy networks could result in the formation of an inert layer on the surface of the materials, which enhances the thermo-oxygen resistance of the epoxy [14]. Chemical copolymerization refers to adding POSS functionalized with different substituents into the epoxy/amine curing system to obtain epoxy resin with a hanging POSS structure. This type of method is the main one used for EP-POSS preparation currently [15,16]. Brus et al. synthesized epoxy/POSS nanomaterials using diglycidyl ether of bisphenol A (DGEBA) and poly(oxypropylene)diamine. Monofunctional POSS as a pendant unit and the aromatic amine 4,4-diaminodiphenylmethane as a curing agent incorporated into DGEBA epoxy crosslinking networks could achieve molecular-level dispersion of pendant POSS in epoxy hybrid networks [17]. Moreover, POSS has great influence in epoxy rubber mixed matrixes. The modulus of resin with pendant POSS structures increased by 1.5 orders of magnitude relative to that of unmodified resin, which was due to the pendant POSS group forming ordered crystalline domains as physical cross-links [18]. Besides this, a small amount of POSS obviously enhanced the flexural strength of EP-POSS nanocomposites, whereas the glass transition temperature (T_g_) value decreased with POSS introduction because there were a certain number of flexible chain segments in the polymer matrix [19]. The addition of a small amount of POSS does not significantly affect the T_g_ value until it is increased to a certain amount; molecular simulations results demonstrated this conclusion [20]. In summary, the above studies showed that POSS has an excellent improvement effect on epoxy resin.

POSS is a cage structure composed of siloxane groups; the Si–O bond energy is 425 kJ/mol, which is higher than that of the C–C bond (345 kJ/mol). The former structure reduces POSS’s surface energy, resulting in a blocking effect on its main chain, which can be used to design hydrophobic material structures [21]. We speculated that introducing the POSS structure into epoxy resin main chains could improve the hydrophobicity of epoxy resin, with an expectation for it to be used in solar panels. To verify our assumption, a simple preparation method was used to synthesize EP-POSS nanocomposites, where POSS was the skeleton structure rather than the hanging group in the EP-POSS. The synthetic route is shown in Figure 1.

## 2. Materials and Methods

### 2.1. Materials

KH 560 was purchased from Chengdu Kelong chemical agent company. Tetramethyl ammonium hydroxide aqueous solution (98%) was purchased from Beijing Couple technology co. Ltd. (Beijing, China). E51 epoxy resin was purchased from Sinopec Group. Other agents were purchased from Sigma-Aldrich (Shanghai, China). All of them were used as received.

### 2.2. The Preparation of EP-POSS

Quantities of 100 mL of isopropanol, 14 g of 4% tetramethylammonium hydroxide aqueous solution, 100 mL of xylene, and 60 g of KH560 were added into a 250 mL three-port flask equipped with a thermometer, a stirrer, and a drip funnel. Then, the KH560 xylene solution was added into the reactor with stirring, completing the addition within 30 min. The solution was then kept under continuous stirring at room temperature for 24 h. The refluxing reaction was carried out in an 80 °C water bath for 2 h, the pH value was adjusted to neutral, and the mixture was stirred at room temperature for 1 h. Then, distillation was carried out to remove the solvent. After washing with acetone and steaming with an alcohol lamp on an evaporation dish twice, a light-yellow viscous liquid was obtained, which was the EP-POSS product. After further drying with the alcohol lamp on an asbestos net, yellow EP-POSS powder was obtained by crushing.

### 2.3. Synthesis of Low-Viscosity POSS (POSS-L) and High-Viscosity POSS (POSS-H)

Distilled water, concentrated hydrochloric acid, anhydrous methanol, anhydrous ethanol, and silane coupling agent (KH560) were added into a 500 mL three-port flask in certain proportions, and the reaction temperature was adjusted to 30 °C for 8 h. After that, rotary evaporation was carried out to further remove the solvent to obtain low-viscosity POSS under a reaction temperature of 35 °C. The above steps were repeated to obtain high-viscosity POSS.

### 2.4. Preparation of the EP-POSS Composite

Certain quantities of epoxy resin and low-viscosity or high-viscosity POSS were mixed and stirred, then mixed evenly with diaminodiphenylsulfone curing agent at room temperature, before pre-polymerizing for 5 h. Then, the mixture was injected into a heated mold, with bubbles pumped in a vacuum oven for about 1 h. Finally, the molded material was cured and shaped according to a curing process of 75 °C for 1 h and 110 °C for a further 1 h. After cooling to room temperature, a further curing process was carried out at 240 °C for 3 h. Finally, standard mechanical test specimens were prepared from the samples.

### 2.5. Synthesis of EP-POSS-SiO_2_ Nano-Composites

The EP-POSS-SiO_2_ nano-composites were prepared by mixing POSS with different viscosity levels and a certain proportion of nano-SiO_2_ in acetone for 10 h, volatilizing the acetone completely at room temperature.

### 2.6. Characterization

Contact angle testing (CA): Contact angle testing (CA) was conducted with 4 μL droplets of water using a Krüss DSA 30 (Krüss Company, Ltd., Hamburg, Germany) device at ambient temperature. WCAs (water contact angles) were measured at least three times for each sample at different spots.

Scanning electron microscopy (SEM)/energy-dispersive spectrometry (EDS): SEM/EDS study was carried out using a Hitachi model S3000H field emission device (Tokyo, Japan), (SM2082EDS Shenzhen Zhixingfeng Technology Co. Ltd., Shenzhen, China) for carbon steel surfaces corroded for 12 h in 1 M HCl solution.

Fourier infrared spectroscopy (FTIR): the structure of the EP-POSS was characterized via attenuated total reflection Fourier transform infrared spectroscopy (ATR-FTIR, Guangzhou zhebo testing technology co., LTD, Thermo Scientific, Guangzhou, China).

X-ray photoelectron spectroscopy (XPS): The elemental composition of the EP-POSS surface was determined via X-ray photoelectron spectroscopy (XPS), conducted in an ultrahigh-vacuum chamber equipped with an X-ray photoelectron spectrometer (SPECS Phoibos 100 MCD analyzer) (Berlin, Germany) and a non-monochromatized Al Kα X-ray source (SPECS XR 50).

Transmission electron microscopy (TEM): TEM was conducted using a Tecnai G2 Spirit Twin12 microscope (Tecnai G2 Spirit Twin 12; FEI Company, Czech Republic). The samples were cut at room temperature using an Leica Ultracut UCT ultramicrotome (Leica Microsystems, Inc., Tokyo, Japan). The 50 nm ultrathin sections were collected on a microscopic grid, covered with a 4 nm carbon layer in order to limit sample damage in the electron beam, and observed in the TEM microscope at 120 kV using bright field imaging.

Thermogravimetric analysis (TGA): Mass loss/temperature curves were obtained using a Perkin-Elmer TGA7 device (TGA4000 (PerkinElmer, USA)). The samples were measured at a heating rate of 10 °C/min. Nitrogen was used as a purge gas.

Si-nuclear magnetic resonance spectroscopy (Si-NMR): Si-NMR was carried out using a Bruke AVANCE 400 MHz (Bruke, Germany) superconducting nuclear magnetic resonance spectrometer, with DMSO as the solvent and TMS as the internal standard. Testing was carried out at 25 °C.

Differential scanning calorimetry (DSC): DSC was performed on a TA Instruments (New Castle, DE, USA) Q20 model using 3–6 mg of sample at a heating rate of 10 °C min^−1^ from 40 to 350 °C under a nitrogen atmosphere.

Atomic force microscopy (AFM): AFM was conducted using a Digital Instruments NanoScope III (Digital Instruments (DI), Santa Barbara, CA, USA) equipped with a J scanner; the scanning range was 150 μm, the force constant was 10 N/m, and the tip radius was 10 nm (NT-MDT).

Flexural performance: the flexural test method followed ASTM D790; the specimen size was 3 × 12.7 × 60 mm^3^ and the test rate was 1 mm/min.

Impact performance: The impact test method followed ASTM D6110; the specimen size was 4 × 12.7 × 125 mm^3^.

## 3. Results

### 3.1. EP-POSS Synthesis Factors

#### 3.1.1. The Relationship between the Solvent Volume Ratio of Xylene and Isopropanol and the Yield of EP-POSS

Figure 2 displays the relationship between the xylene/isopropanol solvent volume ratio and the EP-POSS yield with 4% catalyst at 80 °C, a raw material hydrolysis time of 12 h, and condensation time of 180 min. It can be seen that under the above conditions, the yield of the product increased first and then decreased; the maximum yield was 94% when the volume ratio of xylene and isopropanol was 0.5.

#### 3.1.2. Relationship between Heating Time and EP-POSS Yield

Figure 3 shows the relationship between the heating time and EP-POSS yield under the conditions of 4% catalyst, equal volumes of isopropanol and xylene in the mixed solvent, a raw material hydrolysis time of 12 h, and a heating temperature of 80 °C. It can be seen that under the above conditions, the yield of the product first increased with increasing heating time, then decreased to 90%. The product yield was the highest when the heating time was 180 min. This indicates that extension of the condensation time leads to an increase in product yield. However, when the heating time is too long, it may also cause side reactions. The above experiment results show that the maximum yield can reach 94% under the conditions of 4% catalyst, 1:1 volume ratio of isopropanol to xylene, and condensation time of 180 min at 80 °C.

### 3.2. Surface Structure Composition

The chemical composition of the EP-POSS coating was investigated via FTIR, Si NMR, and XPS measurements. The FTIR and Si NMR spectra are shown in Figure 4. As can be seen in Figure 4, the peaks at 2935 cm^−1^, 1020 cm^−1^, and 850 cm^−1^ are the C–H stretching vibration, the characteristic peak of the C–O ether structure, and the Si–C stretching vibration, respectively. The sharp peak at 1092 cm^−1^ is attributed to the cage-like Si–O–Si opposition stretching vibration. In addition, the characteristic peaks of alcoholic hydroxyl groups at 3200–3500 cm^−1^ disappeared, indicating that the KH 560 was completely consumed. In the Si NMR spectrum of EP-POSS, there is a sharp peak at −83.0 ppm, suggesting that most of the Si elements are in the same chemical environment, and the main target product is a symmetrical cage structure.

The XPS full spectrum and silicon spectrum are shown in Figure 5. The element structure and content data obtained after peak fitting of each element are shown in Table 1. It was found that the EP-POSS hydrophobic coating was composed of C, O, and Si elements. The structural contents of C–Si and Si–O were 7.30% and 0.54%, respectively, which indicates that the Si element exists in the two structures C–Si and Si–O after the reaction, without other by-products. At the same time, the content of the Si–O structure was significantly lower than that of the C–Si structure. This is due to the fact that the cage structure of POSS reduces the detection sensitivity of the Si–O structure.

### 3.3. Microscopic Morphology Analysis

The microscopic morphology of the EP-POSS coating was investigated via FE-SEM (field emission scanning electron microscopy), transmission electron microscopy (TEM), and atomic force microscopy (AFM) measurements. Figure 6a,b shows SEM images of the EP-POSS film. It can be seen that the coating was composed of a large number of particles, and the particle size was mainly distributed in the micron and nanometer scale, approximately 5–100 μm, with some small particles with diameter 100–300 nm forming the micro-nano composite rough structure. This nano-micro structure can greatly improve the hydrophobic performance of the coating, which is consistent with the CA results.

As shown in Figure 6c,d, the EP-POSS was investigated via energy-dispersive X-ray spectroscopy (EDS). The EDS mapping of the EP-POSS hybrid film further confirmed the existence of C, O, and Si elements, so the results of the EDS were consistent with those of the XPS test. Figure 6e,f shows AFM images of the EP-POSS at different magnifications. It shows that the surface of the EP-POSS was composed of an oriented microstructure similar to the cage structure observed in the high-magnification AFM image, suggesting that the POSS structure was within the epoxy resin.

Figure 6g,h shows the TEM diagrams of EP-POSS hybrid material. To facilitate comparison with the background, the TEM spectrum was observed at the edge of the film. The TEM showed that the black spot at about 1 nm was EP-POSS, which was uniformly distributed in the epoxy resin without any large-area agglomeration. The above results prove that EP-POSS achieved molecular dispersion in the epoxy resin and had a good hydrophobic microstructure.

### 3.4. Hydrophobicity and Light Transmittance Testing

In order to verify the hydrophobic performance, contact angle (CA) measurements were carried out. As shown in Figure 7, the contact angle of EP-POSS was 125°, which is higher than those of other epoxy resins with hanging POSS groups and most resins, even PTFE(poly tetra fluoroethylene) [22,23,24,25,26], indicating that EP-POSS has good hydrophobicity and can be used as a hydrophobic material.

Protective coatings for solar panels have higher requirements for light transmittance. If the hydrophobicity is increased, but the light transmittance is reduced, the material does not have application value. The transmittance was tested by examining the UV–vis spectrum. As shown in Figure 8, 60% of visible light passed through an EP-POSS film with a thickness of 200 μm, indicating that POSS was homogeneously dispersed in the epoxy resin matrix at the molecular level. However, the film’s light transmittance of 60% is too low, and is attributed to yellowing of the film during the drying process with an alcohol lamp. This problem will be addressed in future work.

### 3.5. Thermal Performance

The introduction of POSS can usually improve the thermal performance of a material. Considering the higher temperature of the actual use environment, DSC and TGA tests were performed on EP-POSS. The test results are shown in Figure 9. It can be seen from Figure 9 that the initial decomposition temperature of EP-POSS is 354 °C, higher than those of most epoxy resins. This is contributed by the POSS rigid groups in the structure. DSC testing showed that EP-POSS does not undergo a pyrolysis reaction before 354 °C; the first thermal decomposition reaction peak occurred at 406 °C, and then it degraded rapidly. The thermal experimental results show that EP-POSS has higher thermal performance than other commercial epoxy resins and is suitable for high-temperature working conditions.

### 3.6. Mechanical Performance

The prepared EP-POSS nanocomposite has excellent hydrophobicity and light transmittance, but can its mechanical properties meet the requirements of a protective coating? For the seven nanocomposites prepared, the mechanical properties were characterized by bending and impact performance testing. The bending strength and modulus are shown in Figure 10, the impact test results are shown in Figure 11, and all the test results are listed in Table 2.

It can be seen from Figure 10 that the bending strength and modulus of EP-POSS showed a certain improvement compared with those of epoxy resin. This is because the introduction of the POSS rigid group increases the mechanical strength. On the contrary, the introduction of POSS leads to a reduction in impact performance, as seen in Figure 9. After adding rigid SiO_2_ nanoparticles with a size of 30 nm, the bending strength was further improved, and the resulting material could be used as a hydrophobic protective coating. Besides this, the high-viscosity EP-POSS-H had higher bending strength, which is attributed to its higher structural regularity, increasing the crosslinking density to obtain good properties.

In order to test the surface hydrophobicity changes of solar panels in the actual environment, simulation testing was conducted using samples of size 26 mm × 76 mm and a 500 g weight. The test method is shown in Figure 12. The sample was put on the sandpaper with a 500 g weight as the load on top, and then an external force was applied on the sample. It was moved 25 cm in a straight line, then rotated 90° clockwise after each wear to ensure that the sample was operated evenly from four directions. After 200 cycle operations, the contact angle of the sample was still 115°, indicating that the prepared EP-POSS hydrophobic coating has good abrasion resistance and can be used as a protective coating for solar panels.

## 4. Conclusions

EP-POSS nanocomposite with POSS groups in the main chain structure was prepared by a simple synthesis method under acidic conditions, and the yield reached 94%. The contact angle of the EP-POSS was 125°, showing excellent hydrophobicity due to the micro–nano structure of the surface. The surface Si element exists as Si–C bonds, which have a main chain shielding effect and are beneficial to improving the material’s hydrophobicity. At the same time, EP-POSS showed excellent heat resistance: its initial decomposition temperature was 354 °C. The flexural strength increased from 118.9 MPa to 126.9 MPa, but the impact performance decreased from 38.0 MPa to 32.6 MPa. This is because the POSS group in the skeleton structure improves its rigidity and thermal stability. In addition, the POSS group enhances the hardness of EP-POSS, and the flexibility of the epoxy segment improves the abrasion resistance. The contact angle of EP-POSS was 115° after 200 rubbing operations, indicating that it can be used as a protective coating for solar panels.

## Figures and Tables

**Figure 1 nanomaterials-11-00472-f001:**
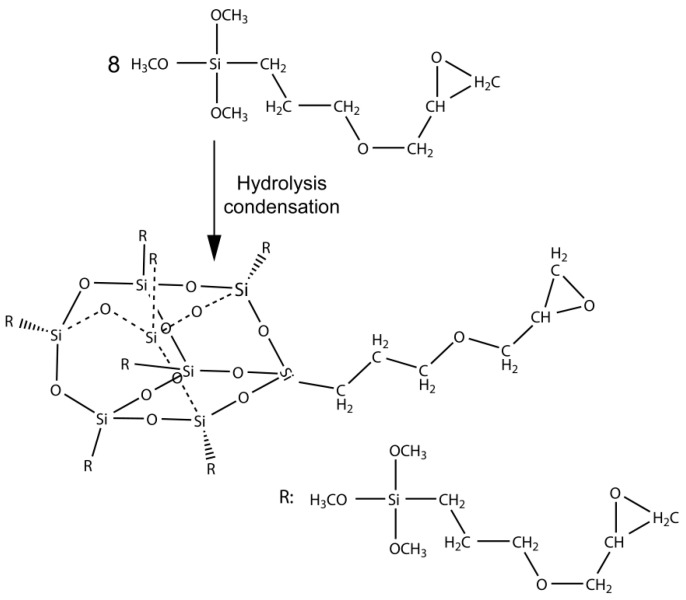
The synthesis route of epoxy caged sesquioxane (EP-POSS).

**Figure 2 nanomaterials-11-00472-f002:**
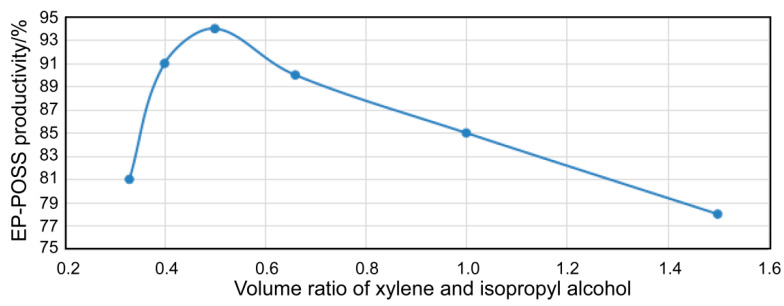
A curve relating the xylene/isopropanol volume ratio with the EP-POSS yield.

**Figure 3 nanomaterials-11-00472-f003:**
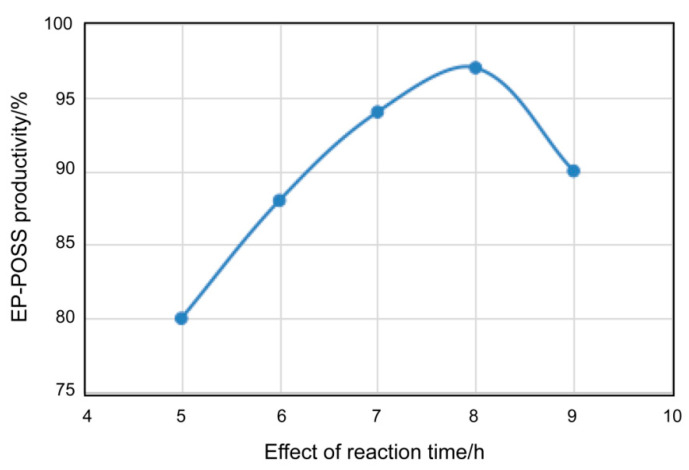
The relationship between heating time and EP-POSS yield.

**Figure 4 nanomaterials-11-00472-f004:**
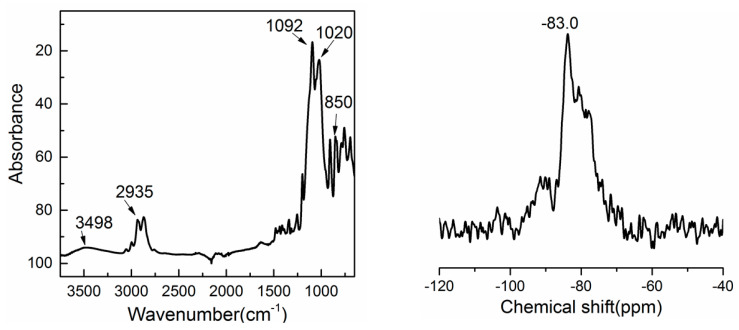
FT-IR (**left**) and Si NMR (**right**) spectra of EP-POSS.

**Figure 5 nanomaterials-11-00472-f005:**
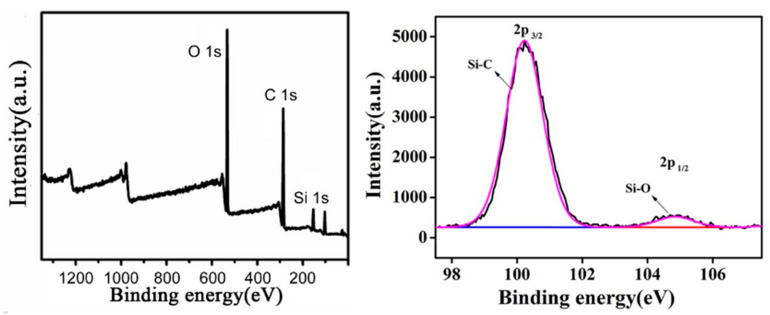
The XPS full spectrum and Si spectrum of EP-POSS.

**Figure 6 nanomaterials-11-00472-f006:**
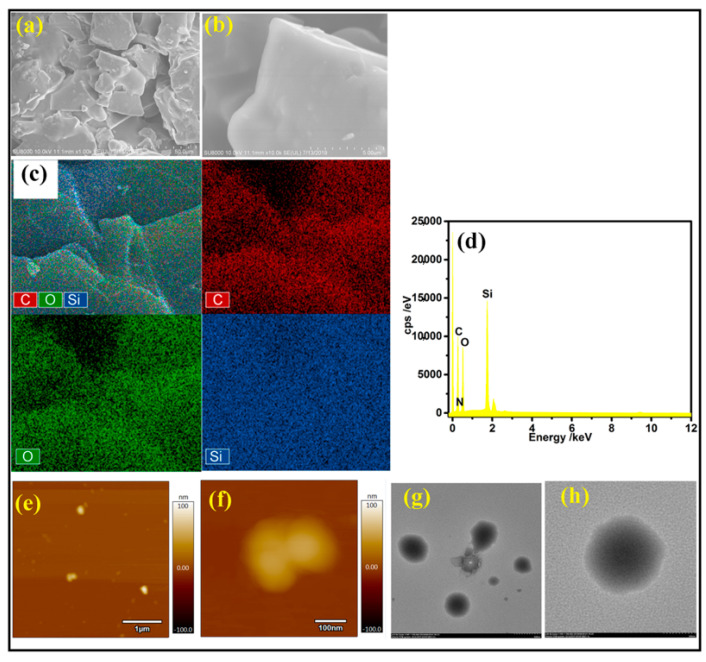
SEM images (**a**,**b**), EDS mapping and EDS elemental contents (**c**,**d**), atomic force microscopy (AFM) images (**e**,**f**), and transmission electron microscopy (TEM) images (**g**,**h**) of EP-POSS.

**Figure 7 nanomaterials-11-00472-f007:**
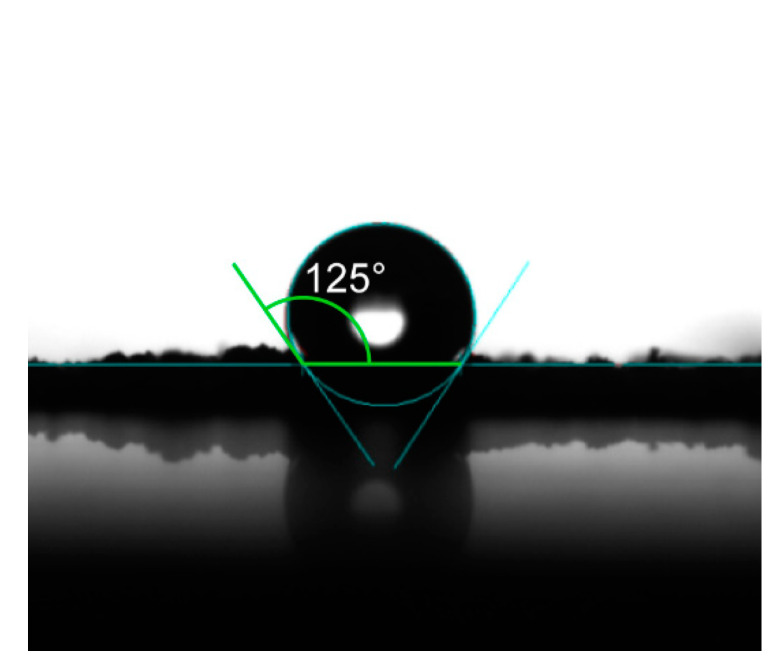
Contact angle measurement of EP-POSS.

**Figure 8 nanomaterials-11-00472-f008:**
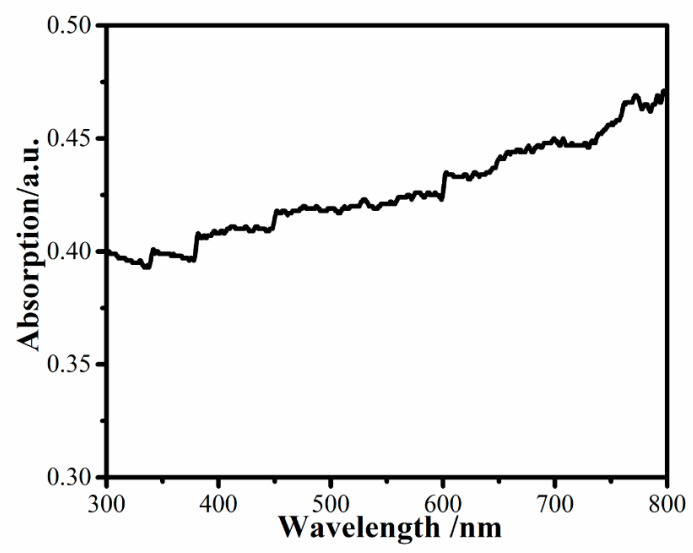
UV–vis spectrum of EP-POSS.

**Figure 9 nanomaterials-11-00472-f009:**
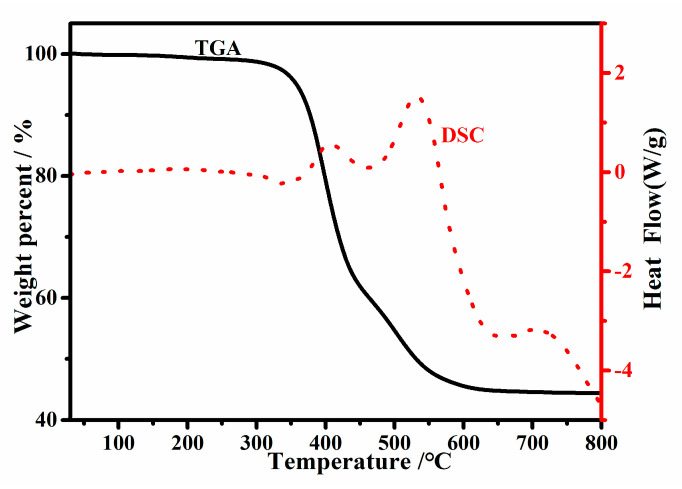
Differential scanning calorimetry (DSC) and thermogravimetric analysis (TGA) curves of EP-POSS.

**Figure 10 nanomaterials-11-00472-f010:**
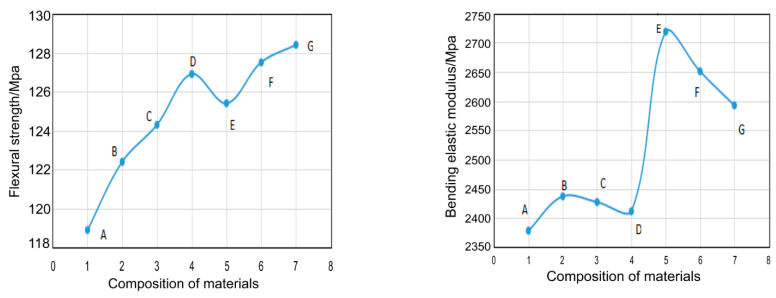
The bending strength (**left**) and modulus (**right**) curves of different material systems for A: EP, B: EP-POSS-L, C: EP-POSS-M, D: EP-POSS-H, E: EP-SiO2-POSS-L, F: EP-SiO2-POSS-M, and G: EP-SiO2-POSS-H.

**Figure 11 nanomaterials-11-00472-f011:**
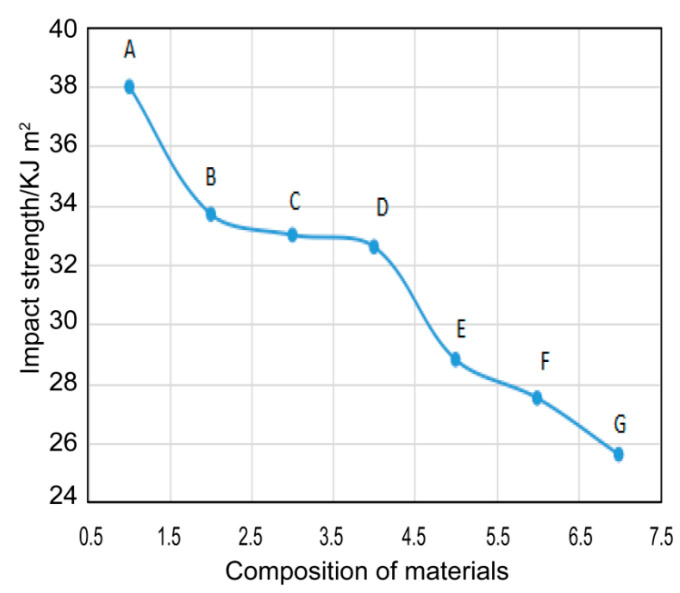
The impact strength curves of the different EP-POSS systems.

**Figure 12 nanomaterials-11-00472-f012:**
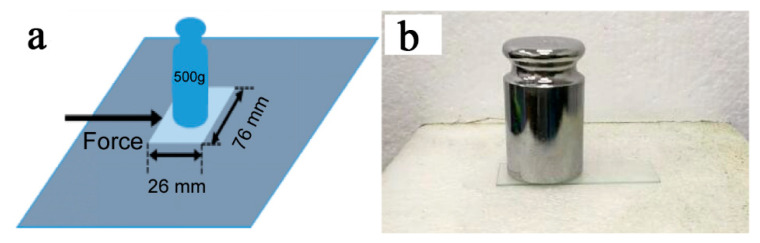
The EP-POSS friction test method.

**Table 1 nanomaterials-11-00472-t001:** As can be seen in Figure 4, the peaks at 2935 cm^−1^, 1020 cm^−1^, and 850 cm^−1^ are the C–H stretching vibration, the characteristic peak of the C–O ether structure, and the Si–C stretching vibration, respectively.

Structure	Si–C	Si–O	C–H	C–O	C–N
Content/%	7.30	0.54	64.61	27.47	0.12

**Table 2 nanomaterials-11-00472-t002:** The mechanical properties of the different EP-POSS systems.

Composition of Materials	Bending Strength (MPa)	Bending Modulus (MPa)	Impact Strength (kJ·m^−2^)
EP	118.9	2378.5	38.0
EP-POSS(L)	122.4	2437.3	33.7
EP-POSS(M)	124.3	2427.4	33.0
EP-POSS(H)	126.9	2411.8	32.6
EP-SiO_2_-POSS(L)	125.4	2718.6	28.8
EP-SiO_2_-POSS(M)	127.5	2650.8	27.5
EP-SiO_2_-POSS(H)	128.4	2592.6	25.6

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
