# Peer review of "Hydrophobic Epoxy Caged Silsesquioxane Film (EP-POSS): Synthesis and Performance Characterization"

_nanomaterials, 2021, doi:10.3390/nano11020472_

Round 1

Reviewer 1 Report

The present study is of great interest if organised properly. The article lacks a reading flow, which can be improved.

  1. The aim of this study should be clearer, in the introduction.
  2. Using POSS for the preparation of nanocomposite is not a new concept. Please highlight the novelty in this work. The novelty should be well compared with other similar reports (if any) and the gap in previous studies should be more prominent in the introduction.
  3. The detailed experimental methods are missing. This will hinder the acceptability of the work due to non-replicability. Eg: the instrument and methods for contact angle study, mechanical analysis (instrument, crosshead speed), FTIR, etc. are missing.
  4. The mechanical analysis data related to the corrosion resistance test are not shown. It is important to support discussions.
  5. The conclusion can be related to the results and aim of the study.

Author Response

Manuscript ID: nanomaterials-998936

Title: Hydrophobic material epoxy cage silsesquioxane film (EP-POSS) Synthesis and performance characterization

Journal: Nanomaterials

Dear Editor,

Thank you very much for your attention and the referee’s evaluation and comments on this manuscript. We have revised the manuscript according to referee’s detailed suggestions. Revised portion are marked in red. Enclosed please find the responses to the referees. We sincerely hope this manuscript will finally meet the requirement of Nanomaterials. Thank you very much for all your help and looking forward to hearing from you soon.

With kindest regards,

Yours sincerely,

Ping Wang

Responds to the reviewer’s comments:

Reviewer #1:

  1. The aim of this study should be clearer, in the introduction.

Reply: Thanks for your valuable advice. We have rewritten the Introduction, and clearly stated the research ideas and purpose of this article.

      2. Using POSS for the preparation of nanocomposite is not a new concept.               Please highlight the novelty in this work. The novelty should be well                     compared with other similar reports (if any) and the gap in previous studies           should be more prominent in the introduction.

Reply: Thanks for your kind suggestion. The novelty of this article compared to other types of research is that we synthesized an EP-POSS with a POSS group in the main chain structure by a simple method, instead of a POSS hanging group. Besides, the EP-POSS has excellent hydrophobicity and is expected to be used in solar panels field. Related content has been revised in the introduction.

  1. The detailed experimental methods are missing. This will hinder the acceptability of the work due to non-replicability. Eg: the instrument and methods for contact angle study, mechanical analysis (instrument, crosshead speed), FTIR, etc. are missing.

Reply: Thanks for your valuable suggestions. We have added the experimental part in the article.

  1. The mechanical analysis data related to the corrosion resistance test are not shown. It is important to support discussions.

Reply: Thanks for your kind suggestion. We have deleted the corrosion resistance test, because we can’t afford detailed test method within 5 days revised time, and this part is not important for our experimental ideas and conclusions.

  1. The conclusion can be related to the results and aim of the study.

Reply: Thanks for your suggestion. We have carefully revised the conclusion part.

Reviewer 2 Report

Dear Authors,

Please check comments in the text, you can find the file attached.

Author Response

Dear Reviewer

Thank you for carefully reviewing this article. We have carefully revised all the questions and errors you raised, including five parts: Abstract, Introduction, Experiment, Results and Conclusion, and all revised paragraphs are marked in red. Please see the attachment. Thanks again for your patience.

Yours sincerely,

Yanhong Fang

Round 2

Reviewer 1 Report

The article has improved a lot after revision. Please include the method for: 

3.6. Mechanical performance

It should be clearly mentioned about the methods for bending and impact performance tests.

Author Response

Thanks for your kind suggestion, mechanical test method has been added in the revised manuscript.

Reviewer 2 Report

Dear Authors, 

you can find my comments in the attached file.  I can see that you made some changes according to the comments I provided. Unfortunately, I think that your work still needs to be improved. Some parts of the manuscript are pretty hard to follow. Also, some sections of the previous version (v1) were clearer and more detailed compared to the actual one (v2), such as the introduction .. I suggest you to work on the two versions you have made and use them to make a complete final manuscript.

I wish you good luck with your work and happy new year.

Kind regards.

Author Response

Thanks for your kind suggestions to improve this article. We have answered your questions in the attached PDF version manuscript, and the relevant modifications are completed in the word version of the manuscript.
Best wish to you have a good health and happy new year!  
